

# Metabolism in a deep hypertrophic aquatic ecosystem with high water-level fluctuations: a decade of records confirms sustained net heterotrophy

Mayrene O. Guimarais-Bermejo[1], Martin Merino-Ibarra[2], Patricia M. Valdespino-Castillo[2], Fermín S. Castillo-Sandoval[2] and Jorge A. Ramírez-Zierold[2]

[1] Posgrado en Ciencias del Mar y Limnología, Universidad Nacional Autónoma de México, Ciudad Universitaria, Mexico

[2] Unidad Académica de Ecología y Biodiversidad Acuática, Instituto de Ciencias del Mar y Limnología, Universidad Nacional Autónoma de México, Ciudad Universitaria, Mexico

Corresponding author
Martin Merino-Ibarra,
mmerino@cmarl.unam.mx

## ABSTRACT

Long-term and seasonal changes in production and respiration were surveyed in the Valle de Bravo reservoir, Mexico, in a period during which high water-level fluctuations occurred (2006–2015). We assessed the community metabolism through oxygen dynamics in this monomictic water-body affected by strong diurnal winds. The multiple-year data series allowed relationships with some environmental drivers to be identified, revealing that water level-fluctuations strongly influenced gross primary production and respiratory rates. Production and respiration changed mainly vertically, clearly in relation to light availability. Gross primary production ranged from 0.15 to 1.26 $gO_2$ $m^{-2}$ $h^{-1}$, respiration rate from $-0.13$ to $-0.83$ $gO_2$ $m^{-2}$ $h^{-1}$ and net primary production from $-0.36$ to 0.66 $gO_2$ $m^{-2}$ $h^{-1}$ within the production layer, which had a mean depth of 5.9 m during the stratification periods and of 6.8 m during the circulations. The greater depth of the mixing layer allowed the consumption of oxygen below the production layer even during the stratifications, when it averaged 10.1 m. Respiration below the production layer ranged from $-0.23$ to $-1.38$ $gO_2$ $m^{-2}$ $h^{-1}$. Vertically integrated metabolic rates (per unit area) showed their greatest variations at the intra-annual scale (stratification-circulation). Gross primary production and Secchi depth decreased as the mean water level decreased between stratification periods. VB is a highly productive ecosystem; its gross primary production averaged 3.60 gC $m^{-2}$ $d^{-1}$ during the 10 years sampled, a rate similar to that of hypertrophic systems. About 45% of this production, an annual average net carbon production of 599 g C $m^{-2}$ $year^{-1}$, was exported to the hypolimnion, but on the average 58% of this net production was recycled through respiration below the production layer. Overall, only 19% of the carbon fixed in VB is buried in the sediments. Total ecosystem respiration rates averaged $-6.89$ gC $m^{-2}$ $d^{-1}$ during 2006–2015, doubling the gross production rates. The reservoir as a whole exhibited a net heterotrophic balance continuously during the decade sampled, which means it has likely been a net carbon source, potentially releasing an average of 3.29 gC $m^{-2}$ $d^{-1}$ to the atmosphere. These results are in accordance with recent findings that tropical eutrophic aquatic ecosystems can be stronger carbon sources than would be extrapolated from temperate systems, and can help guide future reassessments on the contribution of tropical lakes and reservoirs to carbon cycles at

the global scale. Respiration was positively correlated with temperature both for the stratification periods and among the circulations, suggesting that the contribution of C to the atmosphere may increase as the reservoirs and lakes warm up owing to climate change and as their water level is reduced through intensification of their use as water sources.

## INTRODUCTION

Ecosystem-scale metabolic rates represent an integrated measure of the ecosystem functioning in terms of organic matter production and consumption (*Odum, 1971*), and are fundamental metrics of the ecosystem (*Hoellein, Bruesewitz & Richardson, 2013*). These integrated measures are needed to build regional and global balances of important biochemical elements, mainly oxygen and carbon (*Raymond et al., 2013*), which may be estimated through the oxygen dynamics linkage to photosynthesis and aerobic respiration. Yet our assessment of the contribution of epicontinental waters to global biogeochemical cycles is hindered by the variety of systems included (*Lewis, 2011*). For example, it has been determined that 50% to 85% of the oxygen we breathe at planetary scale comes from oceanic primary production, but the global supply from inland waters has not yet been calculated (*Downing, 2014*). The recent re-evaluations of the contribution of epicontinental water bodies to global carbon balance (*Cole et al., 2007*; *Alin & Johnson, 2007*; *Tranvik et al., 2009*; *Lewis, 2011*; *Raymond et al., 2013*; *Downing, 2014*) illustrate the present controversy regarding its magnitude.

One of the main causes of the uncertainty regarding regional and global budgets is the scarcity of studies on carbon fluxes and  community metabolism in tropical lakes and reservoirs (*St. Louis et al., 2000*; *Staehr et al., 2012*; *Sarmento, 2012*; *Raymond et al., 2013*; *Almeida et al., 2016*). *Duarte & Prairie (2005)* pointed out that most rivers and oligo- to mesotrophic lakes are likely carbon emission sources, a result confirmed by *Hoellein, Bruesewitz & Richardson (2013)* who found that 61% of the lakes they compared—mostly temperate—were heterotrophic. Furthermore, recent studies show that in eutrophic tropical water bodies the respiration can override production and can render them as significant sources of carbon emission (*Gupta et al., 2008*; *Almeida et al., 2016*; *Räsänen et al., 2018*). Because eutrophication ranks in first place among the problems faced by limnology, and it is expected to remain as the main issue in aquatic ecosystems in the future (*Downing, 2014*), it is important to study the metabolic balance of tropical eutrophic systems.

Climate change is another very important process affecting water bodies and their role in global cycles (*Kosten et al., 2010*). To address its effects and to better predict future trends, we need long-term monitoring of the metabolism of water bodies to

assess metabolic variability at local and regional levels (*Staehr et al., 2010*; *Sarmento, 2012*; *Solomon et al., 2013*; *Agusti et al., 2017*). Long-term monitoring of whole ecosystems can help identify variations related to their environmental drivers, because they are likely to include significant variations of these drivers without the scaling problem of experimental assessments. Among these drivers, physical features are the least studied in relation to their effects on metabolism, in part due to the complexity of disentangling the simultaneous effects of multiple drivers (*Hoellein, Bruesewitz & Richardson, 2013*; *Coloso, Cole & Pace, 2011*; *Hararuk et al., 2018*). Nutrients are among the main drivers of primary production in oligotrophic and mesotrophic conditions, hindering the identification of the effects of physical features. Because of this, in eutrophic and hypertrophic systems, where nutrient availability is not limiting, the effects of physical drivers may be easier to isolate and quantify (e.g., *Coloso, Cole & Pace, 2011*; *Hararuk et al., 2018*).

Water-level fluctuations are among the drivers of planktonic shifts and ecosystem functioning (*Geraldes & Boavida, 2005*; *Wantzen et al., 2008*; *Mac Donagh, Casco & Claps, 2009*; *Zohary & Ostrovsky, 2011*; *Kolding & Van Zwieten, 2012*; *Valeriano-Riveros et al., 2014*). However, relationships of these fluctuations with metabolic descriptors need clarification; one of their consequences might be the enhancement of boundary mixing events and hypolimnetic entrainment (*Valdespino-Castillo et al., 2014*; *Ramírez-Zierold et al., 2015*), which are aspects even more seldom approached in the tropics (*Zohary & Ostrovsky, 2011*).

Because of this, in the present study we evaluate the long-term (2006–2015) dynamics of the production and respiration rates of a tropical, eutrophic reservoir exposed to high wide fluctuations in the water -level. We aim to identify the variability and trends of its metabolism and the role of physical drivers (water level, temperature and transparency). We also intend to gain insight into the role of this type of system in the carbon cycle, to provide data from tropical systems for national, regional and global scale carbon flux quantifications, and to provide information to improve the efficiency with which epicontinental aquatic systems are managed.

## STUDY AREA

Valle de Bravo (VB) is a high-altitude (1,830 m a.s.l) tropical reservoir in central Mexico (19°21′30″N, 100°11′00″W) that receives water from a 546.9 km$^2$ forested watershed. It is the largest reservoir (18.55 km$^2$ surface area, mean depth 21.1 m, maximum 38.6 m and a storage capacity of $391 \times 10^6$ m$^3$) of the Cutzamala System (comprising seven reservoirs), which provides more than one-third of the water supply to the Mexico City Metropolitan Area (*Ramírez-Zierold et al., 2010*). During the past decade, water extraction for human use has caused wide variations in the water level (*Valdespino-Castillo et al., 2014*; *Ramírez-Zierold et al., 2015*), which has fallen by up to 12 m below the maximum capacity of the reservoir (Fig. 1).

Climate in VB is sub-humid, warm to temperate with pronounced dry (November–May) and rainy (June–October) seasons. Mean monthly temperature ranges from 21.3 °C during May to 15.9 °C in the coldest month (January). The winter minimum varies depending

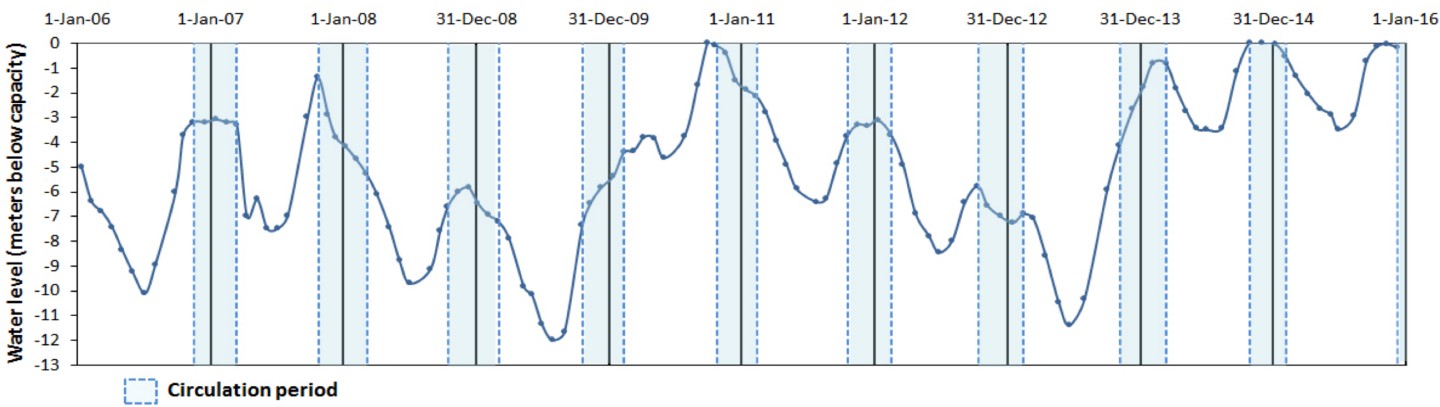

**Figure 1** **Water level fluctuations (meters below capacity) in the Valle de Bravo reservoir from 2006 to 2015.** Blue dotted boxes indicate the circulation periods.

on the number and intensity of cold fronts that reach the reservoir, with historic winter temperatures ranging from 8.1° to 23.4 °C during January. Mean annual precipitation is 836 mm, and mean annual evaporation is 1620 mm (*Ramírez-Zierold et al., 2010*). Strong (7.4 m s$^{-1}$ mean speed) diurnal (12:00–19:00 h) winds that blow along the two main valleys (*Merino-Ibarra et al., 2008*) make VB the most popular inland sailing resort in Mexico.

Because of this climatic pattern, the reservoir behaves as a warm monomictic lake; it remains stratified from March to October and circulates during the winter seasons (November to February). During the stratification period the hypolimnion is anoxic, whereas during mixing the whole water column remains under-saturated (60%) (*Merino-Ibarra et al., 2008*). This is likely due to its eutrophic condition, which is driven by increasing nutrient loads; these have reached up to 116.8 t P year$^{-1}$ and 557.1 t N year$^{-1}$ (*Ramírez-Zierold et al., 2010*). As a result, nutrient concentrations in the reservoir have remained high since 2002 (*Merino-Ibarra et al., 2008*; *Valdespino-Castillo et al., 2014*); during 2002–2015, dissolved inorganic N (DIN) averaged 21.1 µM, and soluble reactive P (SRP) 0.71 µM (*Sacristán-Ramírez, 2017*; *Barjau-Aguilar, 2018*).

These concentrations are well above the limitation thresholds for phytoplankton (*Reynolds, 1999*) and so far no evidence has been found of nutrient limitation of primary production in this eutrophic system (*Merino-Ibarra et al., 2008*; *Valdespino-Castillo et al., 2014*; *Valeriano-Riveros et al., 2014*). High chlorophyll *a* (Chl *a*) concentrations (130–177 mg m$^{-2}$) and small Secchi depth (1.2–2.5 m; *Valdespino-Castillo et al., 2014*) found in VB are consistent with these findings. The plankton community in the reservoir changes in abundance and composition, only loosely in relation to periods of stratification and circulation. During the stratification period, phytoplankton is generally dominated by cyanobacteria, whereas during the circulation period diatoms are more likely to be dominant (*Ramírez-García et al., 2002*; *Valeriano-Riveros et al., 2014*). The zooplankton community is dominated by small-size species, particularly rotifers, as a consequence of cyanobacterial dominance (*Nandini, Merino-Ibarra & Sarma, 2008*), although cladocerans and copepods are also present. Peaks of cladoceran biomass have been observed sometimes

during the circulation period, when diatoms are dominant (*Ramírez-García et al., 2002*) and also during periods when water levels are low (*Jiménez-Contreras et al., 2009*).

Physical processes, mainly mixing events, likely play an important role for nutrient recycling, and plankton composition and metabolism in VB. The strong diurnal winds produce internal waves during the stratification period (*Merino-Ibarra et al., 2008*), and it has been suggested that -in combination with water level fluctuations - these waves enhance boundary-mixing events and hypolimnetic entrainment, particularly during the lower water levels, when the internal waves can interact with the bottom and shoreline (*Ramírez-Zierold et al., 2015*).

In a first analysis of the metabolic variations in VB during the initial year (August 2006–August 2007) of this monitoring, *Valdespino-Castillo et al. (2014)* found that these water level fluctuations affected metabolic rates, particularly respiration, presumably because of their intensification of mixing. Furthermore, they concluded that the effect of mixing hindered the know effect of temperature over respiration when a single stratification-circulation cycle was analyzed. Similarly, in their analysis of the phytoplankton community in VB during 2008–2009, *Valeriano-Riveros et al. (2014)* found peaks of unexpected abundance of diatoms during the stratification periods, which they attributed to nutrient replenishment to the epilimnion due to boundary mixing events and hypolimnetic entrainment driven by the particularly low water levels of VB during 2008–2009 (Fig. 1).

Carbon burial rates have also been assessed in VB using sedimentary records and radiometric methods (*Carnero-Bravo et al., 2014*). These authors found that C flux to the sediments increased significantly after 1991, rising from a previous average of 174 g C $m^{-2}y^{-1}$ to an average of 250 g C $m^{-2}y^{-1}$) between 1993 and 2005.

## MATERIALS AND METHODS

### Environmental variables

Samples were collected and incubated ∼monthly from July 2006 to December 2015. Environmental parameters were determined before and after oxygen incubation, and then averaged. Temperature, dissolved oxygen concentration and its saturation were determined at 1 m vertical intervals down the full water column by means of a multi-parametric probe (Yellow Springs Instruments model 6600). Secchi depth was measured with a standard disk. Chl-*a* concentrations were determined in water samples collected at the 1, 2, 4, 6 and 8 m depths in a Niskin bottle. Samples for Chl-*a* were filtered with 0.45 μm membranes (Millipore, Burlington, MA, USA), extracted with 90% acetone and determined with a spectrophotometer.

### Oxygen dynamics

Oxygen evolution was quantified by incubations of light and dark bottles *in situ* following *Wetzel & Likens (1991)* and *Valdespino-Castillo et al. (2014)*. The oxygen incubations approach to metabolism measurement has some important limitations, mainly that it is very labor intensive because it requires the deployment and recovery of the incubations, it can cause containment artifacts (mainly if the required incubation time is long), and issues related to upscaling if samples are not representative (*Staehr et al., 2012*). In contrast, it has
the advantage of involving a direct measurement of the process, and it does not require a quantification of the air-water oxygen flux (*Staehr et al., 2012*). Because of the strong winds, incubations are better suited for monitoring metabolism in VB than free-water oxygen monitoring because of the impossibility of calculating the strong air-water exchange (*Merino-Ibarra et al., 2008*). This was verified in the analysis of the first year of monitoring (*Valdespino-Castillo et al., 2014*), during which the metabolic rates obtained were one order of magnitude higher than the errors of the method.

Samples and measurements were taken monthly at a central station (cf. *Valeriano-Riveros et al., 2014* for the exact location); this is supported by the verification of horizontal homogeneity in the mixed layer of the reservoir (*Nandini, Merino-Ibarra & Sarma, 2008*) due to intense daytime wind (*Merino-Ibarra et al., 2008*). The bottles were incubated for four to six light-hours at depths of 0, 1, 2, 4, 8, 12, 20, 24 m, and as close to the bottom as possible, depending on the depth of the reservoir at the moment of sampling. At each depth, nine oxygen bottles were filled (three for initial oxygen determination, three for light incubation and three for dark incubation). During the sampling, extreme precautions were taken to completely avoid bubbling that could alter the oxygen, as recommended by *Valdespino-Castillo et al. (2014)*. Dissolved oxygen concentration in each bottle was determined in the laboratory, in triplicate for each sample bottle to minimize and assess error (cf. *Valdespino-Castillo et al., 2014* for further details on the method).

## Metabolism calculations

Gross primary production (GPP), net primary production (NPP), and community respiration ($R$) were calculated using the oxygen change rate in the light and dark bottles, respectively, following *Wetzel & Likens (1991)*, therefore dividing the differences between initial and final oxygen concentrations by the specific incubation time of each set of bottles. Following *Valeriano-Riveros et al. (2014)*, the depth of the production layer ($Z_{pl}$; assumed to end where GPP = 0) was calculated using Secchi depth (SD). To do this, we analyzed the correlation between SD and the GPP = 0 depth in those production profiles in which this point was found in one of the sampled depths. The data fitted better when stratification and circulation data were separated, therefore two correlations (stratification and circulation) were obtained: a coefficient of 4.149 ($n = 26, R^2 = 0.85$) for stratification, and 3.295 ($n = 29, R^2 = 0.90$) for circulation. The corresponding coefficient was multiplied by SD to obtain $Z_{pl}$ in those samplings in which it could not be easily identified in the vertical profile, because GPP = 0 did not concur with one of the sampled depths.

To obtain area-based rates, volumetric rates were integrated over the production layer: the rate from each of the depths sampled was multiplied by the height of the water layer it represented. The same procedure was used when integrating $R$ rates below the production layer ($R_{bpl}$), which were integrated from $Z_{pl}$ to the bottom of the reservoir during circulation, or to the depth where total anoxia was found during stratification. Both respirations ($R_{pl}$, respiration in the production layer) were added to obtain the total respiration ($R_{Total} = R_{pl} + R_{bpl}$) for the full water column.

To calculate diel rates, the hourly production rates were multiplied by the corresponding photoperiod for each sampling date at this latitude. In the case of diel respiration, the

respiration during the night in the production layer was calculated by multiplying the nighttime hours by the dark respiration rate, conservatively estimated as 10% of the GPP rate following *Geider & Osborne (1989)*. Below the production layer, diel respiration was assumed to be constant and was therefore estimated by multiplying $R_{bpl}$ by 24 h. The proportion GPP: $R_{Total}$ in carbon units was used to assess the metabolic balance of the ecosystem as a whole.

Conversion of oxygen rates to carbon rates used the theoretical and most widely used conversion values PQ = 1.3 and RQ = 1.0 (*Gazeau et al., 2005*). The f-ratio (NPP/GPP), *sensu Falkowski et al. (2003)*, was used to assess the fraction of the production that could potentially be exported from the production layer through the sinking of biomass).

## Data analyses

Nutrient data from *Sacristán-Ramírez (2017)* and *Barjau-Aguilar (2018)* were used for preliminary assessment of the relations between nutrient availability and metabolic fluxes. To assess the effect of water-level fluctuations, we calculated the relative lake-level fluctuation index (RLLF) as proposed by *Kolding & Van Zwieten* (*2012*, Eq. (1)). The RLLF is a simple empirical indicator defined as the mean amplitude of the annual or seasonal fluctuations in lake level divided by the mean depth of the lake or reservoir, times 100. We calculated the RLLF both by year and by period (stratification and circulation) to identify the time scale in which water-level fluctuations exert the greatest influence on metabolism of the reservoir.

$$RLLF = \text{mean lake level amplitude/mean depth} * 100. \tag{1}$$

Contour diagrams were constructed in Surfer 11.0.642 (Golden Software, Inc.). The Kriging algorithm with the default linear variogram was used. For the temperature and oxygen diagrams (Figs. 2 and 3) a grid of 2,632 × 108 was used in order to obtain point spacing of approximately one month on the horizontal axis and 1.1 m on the vertical axis. Vertical variation contours of the metabolic rates (Fig. 4) were obtained from monthly averages from the surface to 4 m for an average year, using a search ellipse of four months and 2 m.

## Estimation of error propagation

Errors and their propagation were determined according to classical error propagation theory (*Ku, 1966*). To obtain confidence intervals (CI) around our calculated variables, the standard error (SE) of the mean was converted directly to CI by multiplying with the *t* value for the desired alpha and specific degrees of freedom (*df*) for each case (*Lehrter & Cebrian, 2010*).

Since our calculations required only sums, products and quotients, we used the simplified equations proposed by those authors for simple step-by-step calculations, where $SE_W$ is the SE of a function $W$ calculated from the means of two or more variables (i.e., $W = f\left(\overline{U}, \overline{V}\right)$). For the sum we used Eqs. (2) and (3) for products and quotients:

$$SE_W \sqrt{(aSE_{\overline{U}})^2 + (bSE_{\overline{V}})^2} \tag{2}$$

$$SE_W \sqrt{\left(\frac{SE_{\overline{U}}}{\overline{U}}\right) + \left(\frac{SE_{\overline{V}}}{\overline{V}}\right)^2}. \tag{3}$$

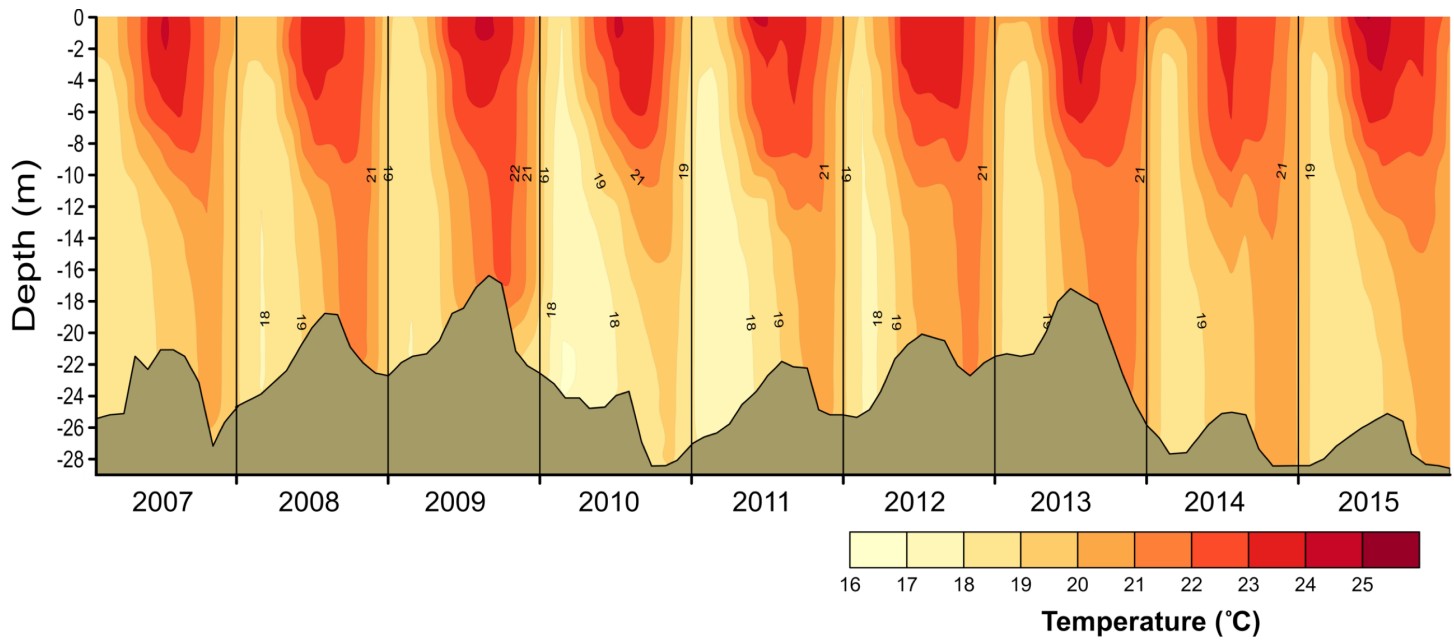

**Figure 2** **Vertical and temporal variation of temperature (°C) in Valle de Bravo reservoir from 2007 to 2015.** Brown shade indicates the bottom below the reservoir.

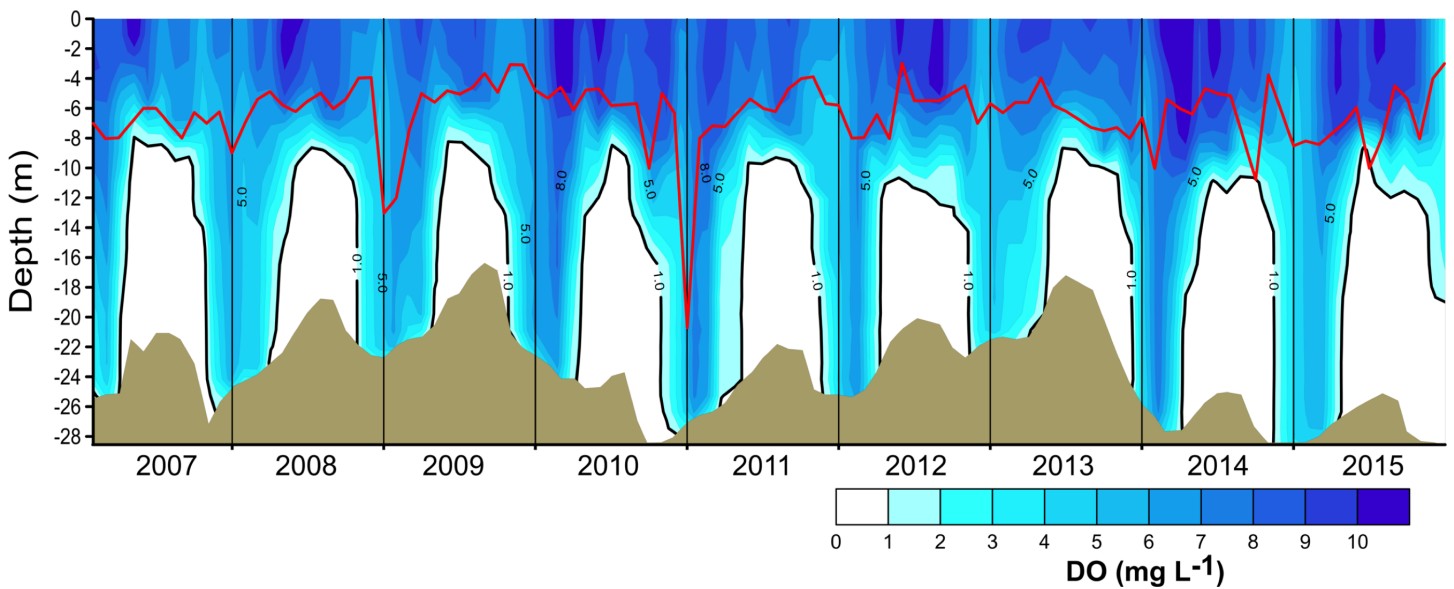

**Figure 3** **Vertical and temporal variation of dissolved oxygen (DO, mg L$^{-1}$) in Valle de Bravo reservoir from 2007 to 2015.** The red line represents depth of the production layer ($Z_{pl}$) and the black line the depth of the mixed layer ($Z_{mix}$), identified by the 1 mg L$^{-1}$ oxygen isoline. Brown shade indicates the bottom below the reservoir.

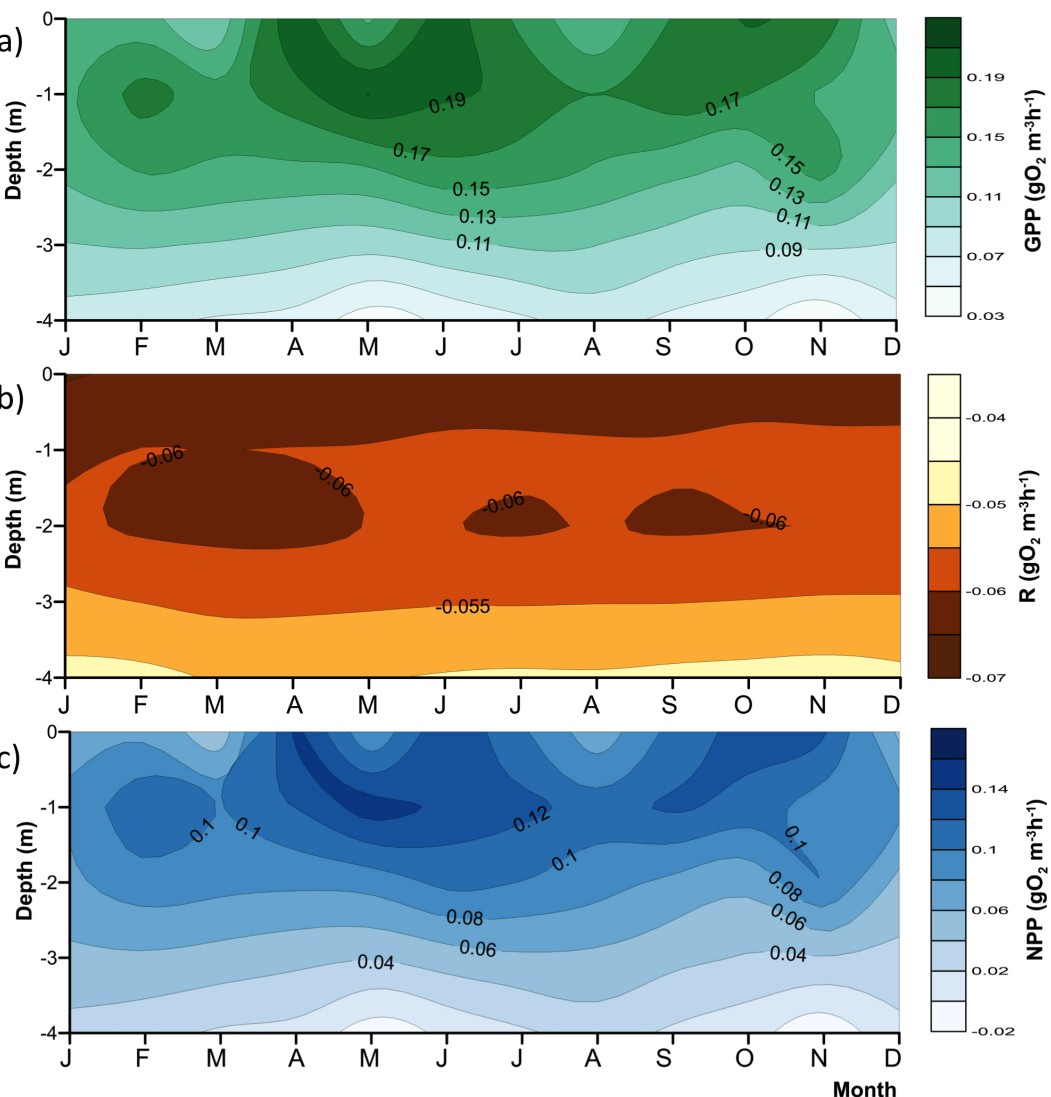

**Figure 4** **Vertical variations of the metabolic rates along the annual cycle within the production layer in Valle de Bravo reservoir: (A) gross primary production (GPP); (B) respiration ($R_{pl}$) and (C) net primary production (NPP).** The figure was constructed with the monthly averages of the data gathered from 2007 to 2015. Letters in the time axis indicate the months (i.e., J, January; F, February, and so on.)

These equations assume that the means involved in the calculation follow a normal distribution, which was checked with the data. Propagation of the degrees of freedom (*df*) from the metabolic rates used Eq. (4) adapted by *Lehrter & Cebrian (2010)* from the Welch-Satterthwaite formula (*Ku, 1966*).

$$df_W = \frac{SE_W}{\sum_{i=1}^{J} \frac{c_i^4 SE_{\overline{W_i}}^4}{df_{\overline{W_i}}}}. \tag{4}$$

**Table 1 Environmental drivers and chlorophyll-a in Valle de Bravo reservoir; averages during the stratification (S), circulation (C) and annually (A) for the 2006–2015 period.**

| Year | Period | | Temperature (°C) | | | Secchi depth (m) | | | Chlorophyll-a ($\mu g\,L^{-1}$) | | | RLLF | | |
| --- | --- | --- | --- | --- | --- | --- | --- | --- | --- | --- | --- | --- | --- | --- |
| | S | C | S | C | A | S | C | A | S | C | A | S | C | A |
| 2006[a] | Aug–Sep | Oct 2006–Mar 2007 | 21.85 | 19.62 | 21.22 | 1.24 | 1.60 | 1.42 | 15.3 | 12.0 | 13.0 | 29.5 | 15.1 | 30.9 |
| 2007 | Apr–Sep | Oct 2007–Mar2 008 | 21.16 | 19.32 | 20.35 | 1.44 | 2.38 | 1.91 | 13.1 | 16.3 | 13.0 | 28.2 | 16.1 | 22.1 |
| 2008 | Apr–Sep | Oct 2008–Mar 2009 | 21.23 | 19.98 | 20.47 | 1.39 | 2.44 | 1.91 | 8.4 | 13.2 | 11.5 | 36.6 | 30.9 | 32.2 |
| 2009 | Apr–Sep | Oct 2009–Feb 2010 | 21.87 | 19.69 | 20.90 | 1.15 | 1.88 | 1.51 | 21.5 | 15.0 | 19.1 | 49.7 | 28.0 | 40.8 |
| 2010 | Mar–Sep | Oct 2010–Feb 2011 | 20.05 | 18.77 | 19.47 | 1.31 | 2.52 | 1.91 | 10.3 | 3.5 | 8.9 | 17.5 | 5.7 | 13.4 |
| 2011 | Mar–Oct | Nov 2011–Feb 2012 | 20.73 | 19.10 | 20.14 | 1.38 | 1.90 | 1.64 | 9.0 | 7.0 | 8.1 | 23.8 | 16.3 | 19.6 |
| 2012 | Mar–Sep | Oct 2012–Mar 2013 | 21.19 | 19.45 | 20.51 | 1.37 | 2.01 | 1.69 | 15.1 | 13.8 | 13.0 | 34.3 | 31.8 | 30.4 |
| 2013 | Apr–Oct | Nov 2013–Mar 2014 | 21.59 | 20.07 | 20.73 | 1.51 | 2.00 | 1.76 | 8.1 | 9.5 | 10.0 | 43.2 | 9.7 | 37.1 |
| 2014 | Apr–Sep | Oct 2014–Feb 2015 | 21.21 | 19.79 | 20.48 | 1.56 | 2.44 | 2.00 | 8.3 | 6.4 | 8.8 | 12.7 | 0.6 | 7.7 |
| 2015 | Mar–Nov | – | 21.18 | – | 20.74 | 1.65 | – | 1.65 | 12.4 | – | 10.4 | 8.9 | – | 6.8 |
| **Mean** | | | **21.21** | **19.53** | **20.50** | **1.40** | **2.13** | **1.74** | **12.2** | **10.7** | **11.6** | **28.4** | **17.1** | **20.5** |

**Notes.**
[a]Sampling began in August.
RLLF = mean lake level amplitude/mean depth ∗ 100 after *Kolding & Van Zwieten (2012)*.

# RESULTS

Because metabolic measurements are done on a volumetric basis and then have to be integrated, first vertically, then for the 24 h period, and finally converted to carbon units, results are here presented sequentially, from the most simple data (i.e., environmental) to the final integrated measurements and their correlation with physical drivers, to address the main question of this paper: how the metabolism of an hypertrophic tropical reservoir behaves in the face of variations in water level and temperature.

## Environmental parameters

Temperature in VB ranged during 2006–2015 between 17.7 and 23.9 °C. Thermal stratification occurred every year, with averages of 22.3 °C during stratification and 19.9 °C in the circulation (Fig. 2). Mean epilimnetic temperatures were always above 22.5 °C during the summer (Jun–Sep, well-established stratification). The lowest temperatures (17.3–19.0 °C) were found in January and February, during the circulation period. These minimal temperatures varied considerably among the years sampled, reaching a minimum during the circulation of 2010–2011 (mean 18.77 °C), and a maximum in the 2013–2014 circulation period (mean 20.07 °C) (Table 1).

The vertical distribution of oxygen found in VB during 2006–2015 (Fig. 3) sharply outlined the monomictical behavior of VB. During the stratification, dissolved oxygen concentrations were close to saturation ($\sim$8 mg L$^{-1}$) from the surface down to $\sim$6 m and thence declined abruptly down to $\sim$10 m depth, below which an anoxic hypolimnion was found. In contrast, during circulation, oxygen concentrations were nearly homogeneous throughout the water column but remained under saturated (50–80 %). Because of this sharp contrast between stratification and circulation, we used the oxycline -and in particular the 1.0 mg L$^{-1}$ oxygen isoline- as a proxy for the depth of the functional mixing

layer ($Z_{mix}$) in VB (Fig. 3), following *Catalan & Rondón (2016)*. $Z_{mix}$ ranged between 8 and 13 m during the stratification periods and deepened to the bottom of the reservoir during most of the circulation periods. Secchi disk transparency varied generally from ~1 m during stratification to ~2 m during circulation (Table 1), although it extraordinarily reached 6.3 m (December 2010). The depth of the production layer ($Z_{pl}$), also shown in Fig. 3, averaged 5.9 m during stratification and 6.8 m during circulation. Chl-*a* concentrations averaged 11.6 $\mu g\ L^{-1}$ overall in VB during the period sampled. The mean concentration during the stratification periods (12.2 $\mu g\ L^{-1}$) was higher than during the circulation periods sampled (10.7 $\mu g\ L^{-1}$, Table 1), but the difference was not statistically significant because variability among periods was high.

## Vertical variations of the metabolic rates (gO$_2$ m$^{-3}$ h$^{-1}$)

As expected, metabolic rates exhibited strong vertical gradients. Figure 4 summarizes these gradients and their annual variations by plotting the mean monthly rates at each depth to obtain the average annual pattern for 2006–2015. Average production (both net and gross) rates were maximal around the 1 m depth (0.09–0.14 gO$_2$ m$^{-3}$ h$^{-1}$ for NPP, and 0.15–0.20 gO$_2$ m$^{-3}$ h$^{-1}$ for GPP) throughout the year, and diminished rapidly and relatively regularly below this depth, clearly in relation to the decrease in light availability. The maximum was higher during the stratification months, from April to July, and showed a secondary maximum around the onset of the circulation period, between September and November. The maximum was smaller and less evident during the central months of the circulation period, December and January, when the vertical variation of production rates was also smaller. The highest variability of production through time was observed in the surface (Fig. 4), where averages ranged from <0.06 to >0.14 gO$_2$ m$^{-3}$ h$^{-1}$ for NPP, and from <0.11 to >0.19 gO$_2$ m$^{-3}$ h$^{-1}$ for GPP. Respiration also had a vertical gradient, and was maximal (around −0.06 gO$_2$ m$^{-3}$ h$^{-1}$, Fig. 4B) at the surface and down to 1 m, presumably as a result of photic stress and photorespiration.

## Vertically integrated metabolic rates (gO$_2$ m$^{-2}$ h$^{-1}$)

Once vertically integrated for the whole production layer, metabolic rates could be graphically depicted for the full period sampled to show the temporal heterogeneity of production and consumption of oxygen (Fig. 5A). The depth of the production layer ($Z_{pl}$), to which the rates were integrated, ranged from 3 to 21 m (Fig. 3). Hourly GPP rates ranged from 0.15 to 1.26 gO$_2$ m$^{-2}$ h$^{-1}$, oxygen consumption from −0.13 to −0.83 gO$_2$ m$^{-2}$ h$^{-1}$ and NPP from −0.36 to 0.66 gO$_2$m$^{-2}$ h$^{-1}$ (Fig. 5A). The three rates showed variability between samplings, but a relative long-term stability.

Oxygen consumption below the production layer ($R_{bpl}$) ranged from −0.23 to −1.38 gO$_2$ m$^{-2}$ h$^{-1}$. $R_{bpl}$ rates during the stratification periods were similar to the production layer values, and the highest rates occurred during the circulation periods (Fig. 5B), when they were integrated from $Z_{pl}$ to the bottom of the reservoir. When the respiration of both layers was considered, the resulting total respiration ($R_{Total}$) for the water column of VB ranged from −0.28 to −2.11 gO$_2$ m$^{-2}$ h$^{-1}$, and the highest rates also occurred during the circulation periods.

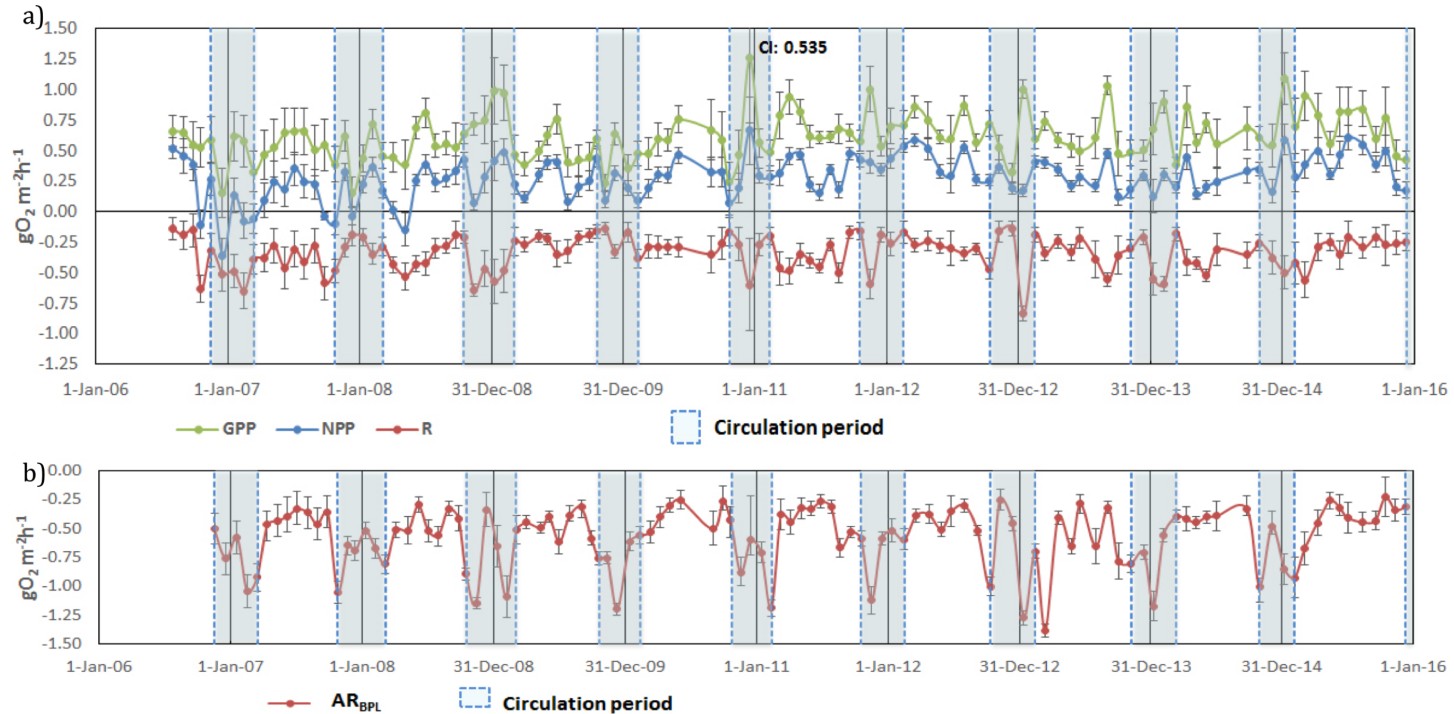

**Figure 5** Temporal variations of: (A) gross primary production (GPP), respiration ($R_{pl}$) and net primary production (NPP) in the production layer and (B) Aerobic respiration below the production layer ($AR_{bpl}$) in Valle de Bravo reservoir from 2007 Error bars indicate the confidence intervals (CI) calculated from the propagated SE and $df$ at $\alpha = 95\%$. Blue dotted boxes indicate the circulation periods.

## Metabolic balance and carbon flux

To assess the metabolic balance and carbon flows, daily rates (24 h-integrated) were considered (Table 2). Although during the daytime the production layer was most of the time autotrophic, when the nighttime respiration was also considered the production layer was sometimes heterotrophic on a daily basis, particularly during the circulation samplings. Moreover, when the respiration from the layer below the production layer ($R_{bpl}$) was also considered, VB as a whole was net heterotrophic throughout the studied period. In fact, the GPP: $R_{Total}$ ratio (in C units) of the reservoir ranged from 0.07 to 0.93 and averaged 0.52 for the full 2006–2015 period (Fig. 6). Although it indicated heterotrophy throughout the full period, the GPP: $R_{Total}$ ratio also showed a very slight long-term trend with an increase of 0.02 y$^{-1}$ (Mann–Kendall trend test, $p < 0.01$).

In terms of carbon fluxes, GPP showed VB to be a very productive system, with an average C fixation of 3.60 gC m$^{-2}$ d$^{-1}$ during the 10 years sampled (Table 2). The NPP averaged 1.64 gC m$^{-2}$ d$^{-1}$, so the potential carbon exportation from the production layer through biomass sinking ($f$ = NPP/GPP) of VB averaged 45% (Table 3) and was higher during the stratification periods (48%) than during the circulation periods (40%). On the annual scale, our measurements of NPP ranged from 258 to 892 g C m$^{-2}$ y$^{-1}$ and averaged 588 g C m$^{-2}$ y$^{-1}$. When compared to the more recent carbon burial rates reported for VB (250 g C m$^{-2}$ y$^{-1}$; *Carnero-Bravo et al., 2014*), the estimated recycling efficiency in the

**Table 2** Mean daily metabolic carbon rates (g C m$^{-2}$ d$^{-1}$) for the production layer and the full ecosystem of VB reservoir during the stratification (S), circulation (C) and annually (A) for the 2006–2015 period.

| | Production layer | | | | | | | | | Full ecosystem | | | | | |
| | GPP | | | $R_{pl}$ | | | NPP | | | $R_{Total}$[c] | | | Net metabolism | | |
| Year | S | C | A | S | C | A | S | C | A | S | C | A | S | C | A |
|---|---|---|---|---|---|---|---|---|---|---|---|---|---|---|---|
| 2006[a] | 3.57 | 2.46 | 3.04 | −1.63 | −2.56 | −1.82 | 1.94 | −0.10 | 1.22 | – | −9.05 | – | – | −6.59 | – |
| 2007 | 3.54 | 2.55 | 3.04 | −2.37 | −1.66 | −2.33 | 1.17 | 0.88 | 0.71 | −5.67 | −8.09 | −7.38 | −2.12 | −5.55 | −4.34 |
| 2008 | 3.54 | 4.40 | 3.47 | −2.31 | −2.56 | −2.15 | 1.22 | 1.84 | 1.31 | −6.05 | −9.79 | −7.12 | −2.51 | −5.38 | −3.66 |
| 2009 | 3.16 | 2.51 | 3.35 | −1.58 | −1.27 | −1.65 | 1.59 | 1.23 | 1.69 | −5.60 | −8.20 | −7.22 | −2.43 | −5.70 | −3.87 |
| 2010 | 3.73 | 3.23 | 3.40 | −1.78 | −1.61 | −1.74 | 1.95 | 1.62 | 1.65 | −5.00 | −8.39 | −5.97 | −1.27 | −5.16 | −2.57 |
| 2011 | 4.40 | 3.83 | 4.03 | −2.38 | −1.48 | −2.05 | 2.02 | 2.34 | 1.98 | −5.78 | −7.62 | −7.03 | −1.38 | −3.79 | −3.01 |
| 2012 | 4.32 | 3.45 | 3.84 | −1.77 | −1.93 | −1.56 | 2.56 | 1.51 | 2.28 | −5.31 | −8.42 | −5.79 | −0.99 | −4.97 | −1.95 |
| 2013 | 3.86 | 3.20 | 3.72 | −2.09 | −1.99 | −2.07 | 1.77 | 1.22 | 1.64 | −7.65 | −8.35 | −8.40 | −3.79 | −5.15 | −4.69 |
| 2014 | 4.21 | 3.98 | 3.81 | −2.50 | −2.10 | −2.32 | 1.70 | 1.89 | 1.49 | −5.78 | −9.31 | −7.10 | −1.57 | −5.33 | −3.29 |
| 2015 | 4.42 | – | 4.31 | −1.79 | – | −1.87 | 2.63 | – | 2.44 | −5.24 | – | −5.99 | −0.82 | – | −1.68 |
| **Mean** | **3.88** | **3.29** | **3.60** | **−2.06** | **−1.83** | **−1.97** | **1.86** | **1.38** | **1.64** | **−5.79** | **−8.58** | **−6.89** | **−1.91** | **−5.29** | **−3.29** |

Notes.
[a] Sampling began in August.
[b] Periods as indicated in Tables 1 and 3
[c] $R_{Total} = (R_{pl} + R_{bpl})$.
GPP, gross primary production; $R_{pl}$, Respiration in the production layer; NPP, net primary production; $R_{Total}$, Aerobic Respiration in the full water column; Net Metabolism calculated as GPP + $R_{Total}$. Positive fluxes imply $CO_2$ assimilation into biomass and negative ones its liberation to the water column.

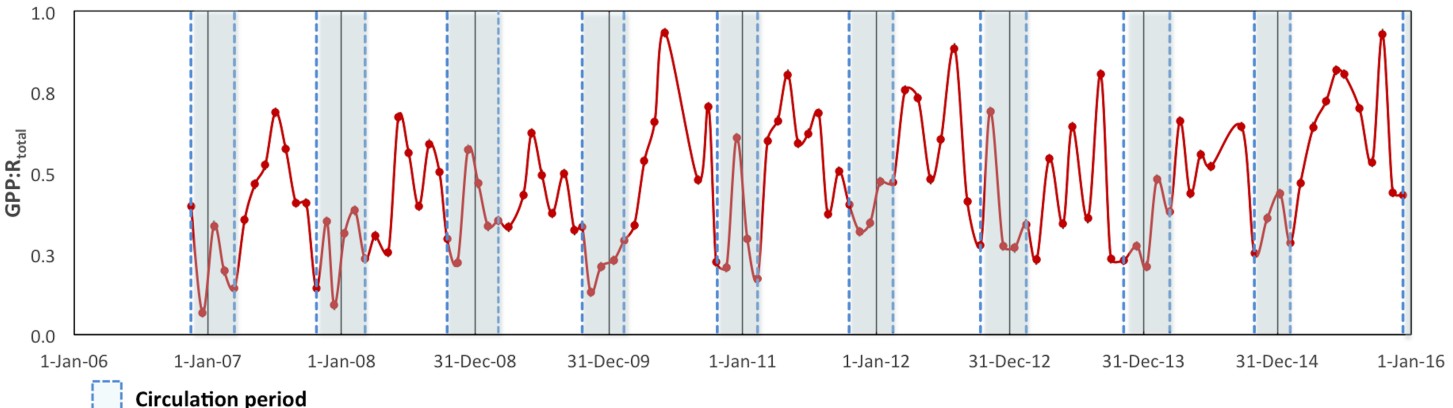

Circulation period

**Figure 6** Ecosystem Metabolic Balance (GPP: $R_{Total}$ quotient in carbon units) for the Valle de Bravo reservoir from November 2007 to December 2015. Blue dotted boxes indicate the circulation periods.

water column of VB would average 58% of the annual NPP and would have ranged from 3% to 72% in the years 2006 to 2015. This means that on the average, only 19% of the total annual 1,344 g C m$^{-2}$ y$^{-1}$ fixed in VB would be sequestered in the sediments, and most of the potential carbon burial would be remineralized back to the water column. Overall, the system's total respiration rate was higher than GPP or NPP, averaging −6.89 gC m$^{-2}$ d$^{-1}$, which in steady state would imply a potential net carbon release of −3.29 gC m$^{-2}$ d$^{-1}$ to the atmosphere on average during 2006–2015.
**Table 3 Fraction of the production (f-ratio) that can potentially be exported from the production layer of Valle de Bravo reservoir during the stratification (S), circulation (C) and annually (A) for the 2006–2015 period.**

| Year | Period | | | | f-ratio |
| --- | --- | --- | --- | --- | --- |
| | S | C | S | C | A |
| 2006[a] | Aug–Sep | Oct 2006–Mar 2007 | 0.54 | −0.04 | 0.40 |
| 2007 | Apr–Sep | Oct 2007–Mar 2008 | 0.33 | 0.35 | 0.23 |
| 2008 | Apr–Sep | Oct 2008–Mar 2009 | 0.35 | 0.42 | 0.38 |
| 2009 | Apr–Sep | Oct 2009–Feb 2010 | 0.50 | 0.49 | 0.51 |
| 2010 | Mar–Sep | Oct 2010–Feb 2011 | 0.52 | 0.50 | 0.49 |
| 2011 | Mar–Oct | Nov 2011–Feb 2012 | 0.46 | 0.61 | 0.49 |
| 2012 | Mar–Sep | Oct 2012–Mar 2013 | 0.59 | 0.44 | 0.59 |
| 2013 | Apr–Oct | Nov 2013–Mar 2014 | 0.46 | 0.38 | 0.44 |
| 2014 | Apr–Sep | Oct 2014–Feb 2015 | 0.40 | 0.47 | 0.39 |
| 2015 | Mar–Nov | – | 0.60 | – | 0.57 |
| **Mean** | | | **0.48** | **0.40** | **0.45** |

Notes.
[a]Sampling began in August.

## Correlations with environmental drivers

Overall, although highly variable among samplings, metabolic rates showed significant differences between the stratification and circulation periods. Total respiration ($R_{Total}$) rates were significantly ($p < 0.01$) higher (1.11 gO$_2$ m$^{-2}$ h$^{-1}$) during circulation than during stratification (0.73 gO$_2$ m$^{-2}$ h$^{-1}$). $R_{bpl}$ rates were also significantly ($p < 0.01$) higher (0.77 gO$_2$ m$^{-2}$ h$^{-1}$) during circulation than during stratification (0.42 gO$_2$ m$^{-2}$ h$^{-1}$), while $R_{pl}$ rates showed a similar pattern, but the differences were not statistically significant ($p < 0.33$).

In contrast, production rates showed the opposite pattern. NPP rates were higher (0.31 gO$_2$ m$^{-2}$ h$^{-1}$) during the stratification than during the circulation (0.26 gO$_2$ m$^{-2}$ h$^{-1}$) and this was also a statistically significant difference ($p < 0.04$). GPP rates were also slightly higher (0.63 gO$_2$ m$^{-2}$ h$^{-1}$) during the stratification than during the circulation (0.60 gO$_2$ m$^{-2}$ h$^{-1}$), but this difference was not statistically significant ($p < 0.20$)

Significant correlations among the rates and the environmental drivers considered were also obtained for each of the two limnological conditions of the reservoir. GPP and NPP did not correlate significantly with temperature, but respiration did correlate positively for both periods. For the stratification periods, $R_{bpl}$ showed a high and significant positive correlation ($r^2 = 0.452$, $p < 0.05$) with temperature. For the circulation periods, when respiration occurs throughout the water column (Fig. 7), $R_{Total}$ also correlated positively ($r^2 = 0.314$, $p < 0.12$) with the mean water column temperature, which varied between years depending on each winter's conditions, as described earlier.

For the stratification periods, significant correlations were also found with the water level index RLLF (higher values of RLLF mean wider water-level fluctuation and lower levels below reservoir capacity). In particular, Secchi depth showed a significant inverse correlation with RLLF ($r^2 = 0.569$, $p < 0.04$). GPP also showed an inverse correlation

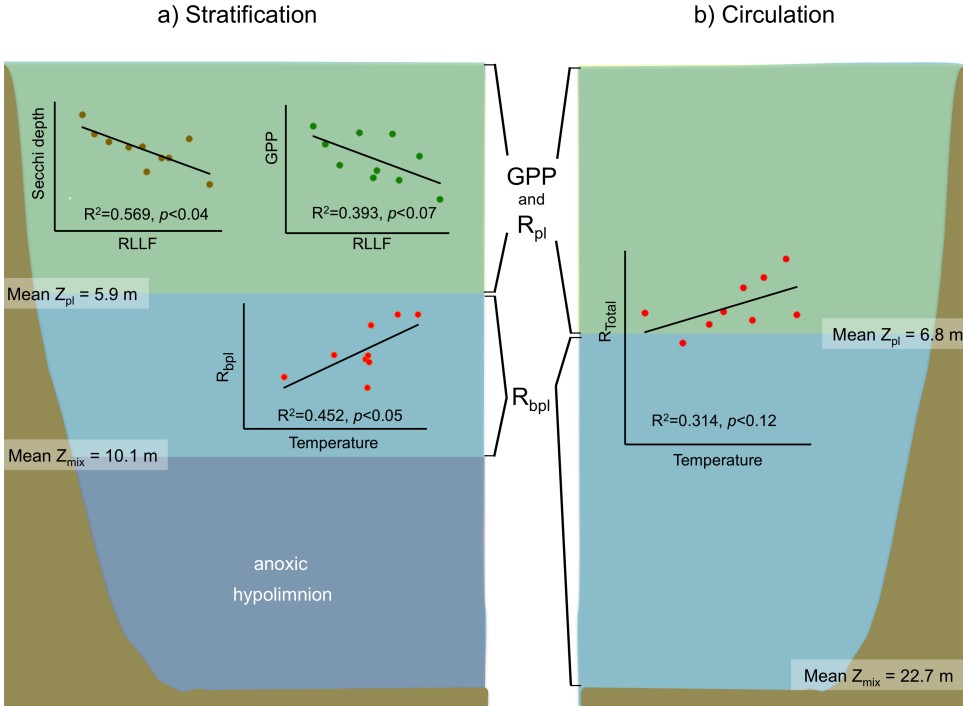

a) Stratification          b) Circulation

**Figure 7 Schematic representation of the functional water layers, the main processes and relations occurring in Valle de Bravo in the stratification and circulation periods during 2006–2015.** All abbreviations as specified in the text. Green shade depicts the production layer. Light blue shade depicts the aphotic layer. Darker blue shade depicts the anoxic layer. Brown shade indicates the bottom below the reservoir.

($r^2 = 0.393$, $p < 0.07$), although the correlations were much less significant during circulation or at the annual scale. Metabolic rates showed significant correlations with nutrients only during the stratification periods. GPP correlated inversely with total P ($r^2 = 0.078$, $p < 0.02$) and with total N ($r^2 = 0.124$, $p < 0.004$). NPP only correlated with total N ($r^2 = 0.129$, $p < 0.003$). $R_{bpl}$ did not correlate with either P or N.

## DISCUSSION

The time series here analyzed reveals considerable impact of physical factors (particularly those associated with fluctuations in water level) on the metabolic rates and the overall heterotrophic behavior of the system. Multiple processes likely occur associated with fluctuations in water level, in a complex interaction where significant vertical changes of metabolism are determinant. Metabolic rates in VB had strong vertical gradients that are likely related to the rapid absorption of light in this eutrophic system, where Secchi disk transparency is generally within 1–2 m. This emphasizes the importance of including detailed vertical sampling in metabolic studies of water bodies, an approach that has been overlooked in many studies (See Table 4). Besides providing insight into vertical changes themselves, detailed vertical measurement of production and respiration rates is important in obtaining realistic and representative area-based integrations, useful for assessing

**Table 4  Metabolism data compiled for previous studies in tropical aquatic systems.**

| Trophic state/System | GPP | | R | | NPP | | P:R | Number of depths sampled | Zpl | Reference |
|---|---|---|---|---|---|---|---|---|---|---|
| | Mean | Range | Mean | Range | Mean | Range | | | (m) | |
| **Oligo-mesotrophic** | | | | | | | | | | |
| Chapala, Mexico | 0.3 | | | | | | | 3 | – | *Lind et al. (1992)* |
| Baringo, Kenya | 0.8 | | | | | | | – | – | *Melack (1976)* |
| Rio Ganjes, India | 1.0 | 0.9–1.0 | | | | | | – | – | *Natarajan (1989)* |
| Titicaca, Peru-Bolivia | 1.1 | | | | | | | 9 | – | *Richerson et al. (1986)* |
| **Eutrophic** | | | | | | | | | | |
| Chad, Chad | 1.3 | | | | | | | – | – | *Melack (1976)* |
| Castanho, Amazona, Brazil | 1.4 | | | | | | | – | – | *Schmidt (1973)* |
| La Mariposa, Venezuela | 1.8 | 0.9–2.6 | | | | | | – | – | *González et al. (2003)* |
| Naivasha, Kenya | 1.9 | 1.5–2.3 | | | | | | – | – | *Melack (1979)* |
| Crescent I. Crater, Kenya | 2.1 | 1.1–3.1 | | | | | | – | – | *Melack (1979)* |
| Chang Jiang, Yangtze, China | 2.4 | 1.1–3.6 | | | | | | – | – | *Liang, Wang & Hu (1988)* |
| Apopka, United States | 2.5 | | | | 1.1 | 1.0–2.0 | | 1 | 1.1 | *Schelske et al. (2003)* |
| Lago Lanao, Filipinas | 2.6 | | | | 1.7 | | | 13 | 15.0 | *Lewis Jr (1974)* |
| Nakuru, Kenya | 2.6 | 0.3–4.9 | | | | | | 12 | – | *Vareschi (1982)* |
| Quebrada Seca, Venezuela | 2.7 | 1.8–3.5 | | | | | | – | – | *González et al. (2003)* |
| Alchichica, Mexico | 2.9 | | 2.0 | | 0.8 | | 1.45 | – | – | *Oseguera et al. (2015)* |
| Victoria, Gulf, Uganda | 3.0 | | | | | | | – | – | *Melack (1976)* |
| **Hypertrophic** | | | | | | | | | | |
| Oloiden, Kenya | 3.1 | 1.6–4.5 | | | | | | – | – | *Allanson et al. (1990)* |
| Kainji, Nigeria | 3.2 | | | | | | | – | – | *Melack (1976)* |
| Poza Yanamalai, India | 3.2 | 1.0–5.4 | | | | | | 1 | – | *Vijayaraghavan (1971)* |
| Tissawewa reservoir, Sri Lanka | 3.3 | 2.6–4.0 | 1.7 | 1.3–2.0 | 1.1 | 0.9–1.2 | 1.55 | 4 | 1.5 | *Amarasinghe & Vijverberg (2002)* |
| Volta, Ghana | 3.3 | | | | | | | – | – | *Melack (1976)* |
| Conway, United States | 3.4 | 0.8–6.0 | 3.8 | 0.8–6.8 | −0.4 | | 0.89 | 8 | – | *Fontaine & Ewel (1981)* |
| Albert, East Africa | 3.5 | | | | | | | – | – | *Melack (1976)* |
| **Valle de Bravo, Mexico** | **3.6** | | **6.9** | | **−3.29** | | **0.52** | **9** | **6.4** | **This study** |
| Victoria offshore, Uganda | 3.6 | | | | | | | – | – | *Melack (1976)* |
| Tanganyika | 3.7 | | | | | | | – | – | *Melack (1976)* |
| McIlwaine, Zimbabwe | 3.8 | 1.6–6.0 | | | | | | – | – | *Allanson et al. (1990)* |
| Pao Canchinche, Venezuela | 3.9 | 1.0–6.8 | 2.8 | 0.3–5.2 | 2.2 | 0.5–3.9 | 1.42 | 4 | 2.2 | *González et al. (2004)* |
| Parakkrama Samudra, Sri Lanka | 4.1 | | | | | | | 7 | 2.1 | *Dokulil, Bauer & Silva (1983)* |
| Shahidullah Hall, Bangladesh | 4.2 | | 3.7 | | | | 1.15 | 10 | – | *Khondker & Kabir (1995)* |
| Bosomtwe, Ghana | 4.7 | | 4.3 | | 0.4 | | 1.10 | 7 | – | *Awortwi (2010)* |
| ES Seridó reservoir, Brazil | 4.9 | | 5.2 | | −0.3 | | 0.94 | – | – | *Almeida et al. (2016)* |
| Estanque Teppakulam, India | 5.0 | 2.0–8.0 | | | | | | 1 | – | *Vijayaraghavan (1971)* |
| George, Uganda | 5.4 | | | | | | | – | – | *Melack (1976)* |
| Victoria offshore, Uganda | 6.8 | | | | | | | – | 13.5 | *Mugidde (1993)* |
| Poza Othakadai, India | 8.7 | 1.5–15.8 | | | | | | 1 | – | *Vijayaraghavan (1971)* |
| Lago Xolotlán, Nicaragua | 9.0 | 6.0 -12.0 | | | | | | 8 | 8 | *Erikson et al. (1998)* |
| Victoria, Pilkington, Uganda | 10.9 | | | | | | | – | 5.0 | *Mugidde (1993)* |

**Notes.**

GPP, gross primary production; R, Respiration; NPP, net primary production; all rates in carbon units (g C m$^{-2}$ d$^{-1}$). Systems are ordered by mean value of GPP. Zpl, depth of the production layer. Dash (−) indicates when the number of sampled levels, or the depth of the production layer, where not specified by the authors. Means and P:R were calculated from the original data when not reported and converted from O$_2$ to C units when necessary.

exchange fluxes of water bodies with the atmosphere and sediments (*Valdespino-Castillo et al., in press*). We estimate that integrated rates can be biased by up to one order of magnitude when based on single-depth measurements.

In VB, the vertical variation of metabolic rates was closely coupled to the vertical changes in limnological parameters and processes. The sharp decrease of production within a few meters is likely caused by the sharp decrease in light availability in this ecosystem where Secchi depth is on average only 1.7 m. The fact that suspended solids in VB have previously been reported to be low (1–6 mg l$^{-1}$) and mainly organic (*Olvera-Viascán, Bravo-Inclán & Sánchez-Chávez, 1998*) suggests that this light limitation is likely an effect mainly of self-shading by the planktonic community itself. Self-shading is very probably due to the very high Chl-a concentrations found in VB, which have been recorded to reach up to 88 mg m$^{-3}$ (*Merino-Ibarra et al., 2008*). This is also supported by the fact that the production layer is always contained (mean $Z_{pl} = 6.2$ m) within the mixed layer (mean $Z_{mix} = 10.1$ m during stratification and 22.7 m during circulation periods; Figs. 3 and 7).

The separation between the depths of these two layers allows the existence of a broad aerobic respiration layer in VB, in which there is not enough light for photosynthesis, but there is still oxygen that is supplied by the mixing processes of the surface layer. During the stratification months, this respiration layer is in the deeper half of the epilimnion, but during circulation periods it extends all the way to the reservoir bottom (Figs. 3 and 7). It is notable that the respiration in this layer outbalances the otherwise autotrophic production layer of this highly eutrophic system. This net heterotrophic character of the ecosystem is likely due to the high nutrient and organic load it receives (*Ramírez-Zierold et al., 2010*), in agreement with the findings of *Solomon et al. (2013)* that both allochthonous organic matter inputs from the watershed, and excess autochthonous production driven by nutrient enrichment, increase background respiration in lakes. Although data on the organic matter content of the water column of VB have not been published, using the dissolved organic nitrogen measurements done by *Barjau-Aguilar (2018)*, we estimate the mean DOC to have been 7.67 mg L$^{-1}$ at VB during the period studied. This value is well above the 5 mg L$^{-1}$, suggested as a threshold for the transition between net autotrophy and net heterotrophy in lakes (*Sobek et al., 2007*), so it is consistent with the net heterotrophy we find in VB. Such an amount of organic matter is probably a factor in the oxygen under-saturation found in VB during the circulation periods. Because the organic load to VB is high (*Ramírez-Zierold et al., 2010*), allochthonous organic matter may be causing a background respiration (*Solomon et al., 2013*) high enough to override the supply of oxygen by mixing.

## Limnological processes

The vertical distribution of temperature in VB found during 2006–2015 confirms the continuity of a monomictic behavior of the reservoir, as found before 2006 (*Merino-Ibarra et al., 2008*; *Ramírez-Zierold et al., 2010*) in spite of the drastic water-level decreases that occurred during this period. Functional thermal stratification occurred every year, although the thermocline gradient was small ($\leq 0.4\,°C\ m^{-1}$), as expected for tropical lakes (*Catalan & Rondón, 2016*). Hypolimnetic temperatures increased in VB during each stratification

period during 2006–2015, confirming the persistence of this trend in VB during the stratification, as reported for previous years (*Merino-Ibarra et al., 2008*; *Ramírez-Zierold et al., 2010*). Maximum temperatures in the hypolimnion occurred during the stratification periods when the water level of the reservoir was lowest (i.e., in 2008, 2009 and 2013, Fig. 2), supporting the suggestion (*Ramírez-Zierold et al., 2015*) that the rate of hypolimnetic warming is related with the magnitude of the water level decrease. Additionally, the 2006–2015 time series here described also shows the interannual variability of temperature in VB, particularly during the circulation periods, when the full water column is affected by heat exchange through the surface, and the water temperature is likely to be directly related to the coldness of each winter.

Oxygen vertical distribution was consistent with the temperature distribution and the monomictic behavior confirmed for VB during the decade sampled. As suggested for tropical eutrophic systems (*Catalan & Rondón, 2016*), we find that oxygen is a better descriptor of the limnological cycle of VB than is temperature. The mainly anoxic hypolimnions (Fig. 3) offer a clear representation of the system's monomictic behavior and its sustained eutrophic condition. Therefore, we concur with *Catalan & Rondón (2016)* that this possibility should be considered for other tropical eutrophic systems, where vertical variations of oxygen are sharper than vertical variations of temperature.

Secchi depth was also a simple but very useful tool to extend the assessment of the vertical range of production ($Z_{pl}$) throughout the full sampled period (Fig. 3). In particular, correlations between SD and GPP = 0 allowed the depth where GPP equals zero to be modeled; this was frequently a depth where experimental incubations were not performed, in spite of the effort we applied to resolve vertical variations through incubating at up to 8 different depths. Hence, the use of Secchi depth to estimate $Z_{pl}$ can be a useful tool to manage large datasets containing vertical assessments of GPP and $R$, and to improve their yield of integrated rates. It is notable that among the metabolic studies in tropical lakes or reservoirs summarized in Table 4, fewer than half report on the measurement of metabolism at multiple water depths, and only one in four reports on the determination of the depth of the production layer. This demonstrates the need for studies that deal in detail with vertical variations of the metabolic rates and therefore offer reliable integrated rates.

### Effects of temperature and water-level fluctuations on metabolic rates

Although temperature is the ultimate driver of metabolic rates at organism level, at the ecosystem scale its effect can be hindered by the many other changes that can occur simultaneously. In particular, (unlike marine systems, cf. *Downing, 2014*), monomictic inland systems exhibit contrasting physicochemical conditions coupled to the seasonal variations of temperature, which are mainly dependent on the variable intensity and spatial extent of mixing processes. These conditions include: the formation of an anoxic hypolimnion where aerobic respiration cannot occur; the transport of plankton across light boundaries during full circulation; and the exposure of reduced compounds to oxidizing conditions and/or the transport of nutrients associated with boundary mixing and hypolimnetic entrainment (*Ramírez-Zierold et al., 2010*; *Ramírez-Zierold et al., 2015*).

All of these processes and conditions can affect production and respiration rates, hindering the effect of concomitant temperature changes. This apparently was the case in the initial assessment of *Valdespino-Castillo et al. (2014)*, who did not find a significant positive correlation between respiration and temperature in tests of 2006–2007 stratification data, or even of annual data for VB. *Valdespino-Castillo et al. (2014)* attributed this to the dominance of the effect of mixing on respiration over that of temperature when stratification and circulation data were assessed together. In contrast, the long-term data now reported reveal a positive correlation of respiration with temperature when circulation and stratification periods were analyzed separately. In the case of the circulations, having data from a series of years allowed enough thermal variability to reveal the expected direct effect of temperature on metabolic rates, and particularly on respiration. Additionally, this expected correlation could also be identified in the case of the stratifications within the lower part of the epilimnion (Fig. 7), where the effects of daily wind mixing would be smaller than in the rest of the epilimnion.

Because of the wide water-level fluctuations in VB during 2006–2015, this data set allowed assessment of the effects of these variations, parameterized here as RLLF. Water level decreases may increase the frequency and intensity of boundary mixing events and hypolimnetic entrainment during stratification in VB (*Ramírez-Zierold et al., 2010*; *Ramírez-Zierold et al., 2015*). In turn, it has also been shown by *Valeriano-Riveros et al. (2014)* that increased boundary mixing favors diatoms over noxious cyanobacteria (Nostocales), owing both to mixing itself and to the nutrient inputs involved. This would mean that water-level fluctuations could affect the food web through changes in mixing. In fact, *Jiménez-Contreras (2009)* also found significant variation in the composition of the zooplankton community, observing that the dominance of rotifers over cladocerans inverted during low-level periods, changing from a short microbial loop to a longer food web. Hence, the impact of water-level fluctuations on metabolic rates likely involves its effect over multiple processes, including mixing itself, changes in the planktonic food web and an increase in nutrient availability to the surface layer.

In our 2006–2015 metabolic data set, where ten stratifications with different water levels could now be compared, both Secchi depth and GPP decreased significantly as a function of the RLLF. The decrease of Secchi depth could be due to the increase in nutrient supply to the surface layer through hypolimnetic entrainment and boundary mixing, where nutrients might be limiting the expansion of phytoplankton. That there may be a certain degree of nitrogen limitation during the stratification in VB was suggested by *Valeriano-Riveros et al. (2014)* and this view is also supported by the low but significant positive correlation we obtained between NPP and TN.

Another process that could be causing decreased Secchi depth and GPP during periods of lower water-levels could be resuspension of fine sediment along the new shorelines; sediments that had settled under conditions of higher water levels would be increasingly exposed as water levels fell. The energy needed for this is available at VB, where strong winds blow daily, but the relative abundance of the sediments suspended in the water column under different water-level conditions still needs to be measured to verify the importance of this possibility.

In either case, these results are consistent with those previously found by *Valdespino-Castillo et al. (2014)* who concluded that sharp water level decreases may shift the community metabolism from autotrophy towards heterotrophy. Altogether, this pattern is also important to direct the needed reassessment of the contribution of reservoirs and lakes to the global carbon cycle (*Cole et al., 2007*; *Tranvik et al., 2009*; *Lewis, 2011*; *Raymond et al., 2013*; *Downing, 2014*), because it shows that lowering of the level would cause a decrease in the net carbon sequestration of deep stratified systems.

## Metabolic balance, spatial and temporal long-term trends

The high GPP found in VB throughout the decade confirms that it has remained a very productive system, in spite of the wide water-level fluctuations and other changes. In fact, its C fixation rates are higher that those found during the initial assessment of 2006–2007 and are now within the range (2.7–5.0 gC m$^{-2}$ d$^{-1}$) of other tropical epicontinental water bodies considered hypertrophic (e.g., Oloiden in Kenya, Lake Conway in Florida, McIlwaine in Zimbabwe, Pao Caniche in Venezuela, and Shahidullah Hall in Bangladesh, Table 4), but not yet as high as, for example, the highly hypertrophic Lake Xolotlán in Nicaragua. Hence, our results in terms of primary production indicate that VB would now be more accurately classified as hypertrophic than as eutrophic.

Although productivity in VB is high, and nearly half of it is net production -which can be exported from the production layer to the sediments and become sequestered C- our respiration data and the comparison with net C burial assessed through independent radiometric methods of *Carnero-Bravo et al. (2014)* indicate that most of this carbon is remineralized and only about 40% of the NPP is permanently buried in the sediments of this ecosystem.

Furthermore, because the total respiration in this ecosystem is also very high -nearly doubling GPP rates- the system has a net heterotrophic metabolism. These results are consistent with the findings of *Gupta et al. (2008)*, *Almeida et al. (2016)* and *Räsänen et al. (2018)* and confirm—now with a long-term data set—that tropical hypertrophic reservoirs can be highly productive and simultaneously be important sources of atmospheric emissions; these emissions may be larger than the burial of organic carbon in their sediments, because of high rates of mineralization in the water column and sediments, as occurs in VB. These findings indicate that the pattern found for temperate systems by *Duarte & Prairie (2005)* and *Hoellein, Bruesewitz & Richardson (2013)* may also take place in eutrophic reservoirs. More metabolic studies on tropical systems that include detailed measurements of vertical variation and respiration are needed, to reassess the contribution of epicontinental water bodies to global carbon balance (*Cole et al., 2007*; *Alin & Johnson, 2007*; *Tranvik et al., 2009*; *Lewis, 2011*; *Raymond et al., 2013*; *Downing, 2014*).

It is expected that long-term data will be key to assessing metabolic variability (*Staehr et al., 2010*; *Sarmento, 2012*; *Solomon et al., 2013*) in the possible scenarios of climate change (*Kosten et al., 2010*). Our results show that a decade of metabolic records can be enough to start identifying trends, as recently found by *Agusti et al. (2017)*. Furthermore, the correlations found here with environmental drivers allow the exercise of simple predictions for VB that can inform global expectations. For instance, our results indicate that for each

1 °C increase in the lake temperature its respiration and C emission could further increase by 0.4–0.9 gC m$^{-2}$d$^{-1}$; this is consistent with the findings of *Kosten et al. (2010)* on CO$_2$ emission from lakes, and suggests one of the negative feedbacks between climate change and eutrophication that could be expected in heterotrophic tropical systems. Similarly, our results on the inverse relationship between GPP and water level decrease (RLLF) also demonstrate that the water deficiency expected for the latitude of VB—if coupled with the intensification of its use as a source of fresh water—could enhance its role as a carbon source, a process that may also occur in the numerous water reservoirs that will be more intensively used throughout similar latitudes.

## CONCLUSIONS

Production and respiration records for VB over the course of a decade show that high respiration of eutrophic tropical reservoirs can surpass their high production and carbon burial rates, and therefore these reservoirs likely act as important atmospheric carbon sources. Temperature and water level variations significantly affect metabolic rates in VB. Mixing, food web changes and nutrient limitation likely play a role that needs to be further investigated. The metabolism of more tropical systems must be studied, in order to improve global budgets, and to build more long-term series to support the prediction of future trends. Our results point to an increase in net heterotrophy of deep eutrophic reservoirs as temperatures increase and as their water levels fluctuate in response to climate change and increased exploitation of their water for human use.

## ACKNOWLEDGEMENTS

This paper is part of the requirements for the PhD degree of M.O. Guimarais-Bermejo. We acknowledge field support from lake users, mainly ProValle A.C. Club Náutico Avandaro and Club Náutico Marina Azul, Gloria Vilaclara and students of the Laboratory of Aquatic Biogeochemistry. We thank Adan Zuñiga for the processing of the Surfer figures, Ann Grant and M. Macek for comments on the manuscript.

### Funding

This work was accomplished with financial support from research projects funded by UNAM-PAPIIT-IN207702, SEMARNAT-CONACYT C01-1125 to M. Merino-Ibarra and by UNAM-PAPIIT IN2089107-3 to J. Carmona-Jiménez. CONACYT awarded a PhD scholarship to M.O. Guimarais-Bermejo, and Idea Wild provided instrumental support. The funders had no role in study design, data collection and analysis, decision to publish, or preparation of the manuscript.

### Grant Disclosures

The following grant information was disclosed by the authors:
Research Projects: UNAM-PAPIIT-IN207702, SEMARNAT-CONACYT C01-1125.

UNAM-PAPIIT: IN2089107-3.
Idea Wild.

## Competing Interests

The authors declare there are no competing interests.

## Author Contributions

- Mayrene O. Guimarais-Bermejo performed the experiments, analyzed the data, prepared figures and/or tables, authored or reviewed drafts of the paper, approved the final draft.
- Martin Merino-Ibarra conceived and designed the experiments, performed the experiments, analyzed the data, contributed reagents/materials/analysis tools, prepared figures and/or tables, authored or reviewed drafts of the paper, approved the final draft.
- Patricia M. Valdespino-Castillo conceived and designed the experiments, performed the experiments, prepared figures and/or tables, authored or reviewed drafts of the paper, approved the final draft.
- Fermín S. Castillo-Sandoval performed the experiments, contributed reagents/materials/analysis tools, authored or reviewed drafts of the paper, approved the final draft, field work and Chemical Analysis.
- Jorge A. Ramírez-Zierold performed the experiments, authored or reviewed drafts of the paper, approved the final draft, field work.

## Data Availability

The raw data are provided as Data S1.

## Supplemental Information

Supplemental information for this article can be found online at http://dx.doi.org/10.7717/peerj.5205#supplemental-information.

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
