# Peer review of "Metabolism in a deep hypertrophic aquatic ecosystem with high water-level fluctuations: a decade of records confirms sustained net heterotrophy"

_PeerJ, doi:10.7717/peerj.5205_

## Round 0.1 · original submission · Major Revisions

Thank you for submitting your interesting study to PeerJ. I have now received reviews of your article from two outside readers, appended below. These reviews indicate that while your article potentially presents an important contribution to the field, revisions are necessary before your article could be accepted for publication.

In your revisions, please give special consideration to: 1) the results section need to be restructured according the main focus of the study; 2) make sure that there is a match between what stated in text and shown in figures; 3) please, provide a deep analysis on the controlling factors of long-term aquatic metabolism, looking for other variables (climate, chemistry, plankton, etc) explaining the remaining variability (if data on chlorophyll-a and dissolved organic carbon are available, it would be great to include them in order to improve the results interpretation); and 4) the discussion section need to be shortened removing obvious eplanations. In addition, during your revision, please make sure that all issues highlighted by reviewer#2 have been answered.

Best regards,

Salva
* * *
Dr. Salvador Sánchez-Carrillo
National Museum of Natural Sciences-CSIC
Madrid, Spain

Reviewer 1 ·

Basic reporting

1.Introduction

The importance of the work is clearly stated on the introduction. It is based on a lack of data and understanding the dynamics of tropical lakes and reservoirs towards a changing world. However, the introduction became too generic in addressing the objective of the research and left out important technical achievements in environmental data (i.e HFM data, hydrobiology…). I would suggest a deepening on this part. Quantity and quality of the references are good. Considering it is an introduction of a tropical study, the paper should give more emphasis to that (including the inclusion of previous works performed in such regions). Thermocline should be addressed.

2.Study Area

The description of the study area is incomplete for the understanding of a layperson. I suggest a better development on the climate description and overall information of the biological system. It is well known that the dynamics of the food web plays an important role on the primary production (see, Carpenter et al. studies in Lake Paul and Peter for instance).

3.Methods

The evaluation of the environmental variables is key for this work. However, the description of how data was granted is severely compromised by the lack of information. The selected methods are biased, but well accepted when other technologies are not available. Nevertheless, in this case, a development paragraph on the limitations of the methodology would be necessary

4.Figures

Please review the figure axis and especially the legends. They must be auto-explained. Figures are non-comprehensives.

5.Results

Results are too fragmented and there is no previous introduction of the analysis that would be done. The work lost its focus and the results became scattered, without a clear research question to be answered. The title suggests that the research question would be: how does a hypertrophic lake behave in face of high-level fluctuation? The way in which the paper was written does not help in logically addressing this question. I would suggest a dramatic re-structuration of the paper based on the clear and precise understanding of the research question. The content is plural and the data can be valuable if well explored.

There was a weak association between what is stated in the text and what is shown on the figures. This applies either for textual expression of the results as extraction of results from the data.

6.Discussion

The discussion is shallow, long, fragmented and brings to the table neither good references, comparative works, discussion of the data in face of other studies and so on. When the work loses the track on the middle it often ends like this. It is not that the information has no value. It just does not hit the point.

Experimental design

Experimental design is impossible to be evaluated due to the lack of information on how data was granted. The manuscript does not stand by itself.

The research area seemed to be well defined in the beginning of the manuscript but missed the track over it. So forth, failed to identify the knowledge gap and work the results towards it. Simply stating that such study is needed in a certain area does not fulfill the expectations of a scientific manuscript.

Validity of the findings

Results are too fragmented and there is no previous introduction of the analysis that would be done. The work lost its focus and the results became scattered, without a clear research question to be answered. The title suggests that the research question would be: how does a hypertrophic lake behave in face of high-level fluctuation? The way in which the paper was written does not help in logically addressing this question. I would suggest a dramatic re-structuration of the paper based on the clear and precise understanding of the research question. The content is plural and the data can be valuable if well explored.

There was a weak association between what is stated in the text and what is shown on the figures. This applies either for textual expression of the results as extraction of results from the data.

Additional comments

Please, have a deeper look on the paper. It has points that are unacceptable. The overall value and research were alright but there are many points of the manuscript that do not agree within itself. The study itself is important and such research shall always be motivated. Unfortunately, at this time the manuscript needs major revision.

The reviewer hereby states its lack of relevance regarding towards background of the researchers involved on this manuscript. The evaluation of the manuscript was based only and exclusively on the quality of the information and writing content, fully disregarding any consideration related to the country of origin, Institute of research or international prestige of the responsible Principal Investigator.

Reviewer 2 ·

Basic reporting

See my general comments.

Experimental design

See my general comments.

Validity of the findings

See my general comments.

Additional comments

This is an interesting study on long-term aquatic metabolism in a subtropical environment. Since there are too few instances of such studies worldwide, it is advisable to publish it. However, there are some hints and topics to be considered by authors before acceptance. They assume that metabolism of Valle del Bravo reservoir is not affected by chemical constraints due to its hypertrophic nature, but they do not offer any proof about. Instead, they make their analysis of controlling factors rely on purely physical features, the most important of which is fluctuating water level, with water temperature and transparency as other, lower significant factors. A proof that all these are not the one and only explanation of metabolic dynamics is that they do not amount to 100% of explained variability of GPP, NPP or Respiration (Lines 272-289). Therefore, they must search for other factors explaining the remaining variability, and the obvious candidates would be climate, the chemical environment and the herbivore plankton. Regrettably, nothing is said about. This undermines a study that is otherwise very interesting and would be welcome by aquatic ecologists worldwide if properly addressing controlling factors of long-term metabolism.
My recommendation is to accept this manuscript for publication, providing that authors thoroughly address this point. Other minor queries and suggestions are the following:
1st) Lines 62-64. “Furthermore, recent studies…”, please give more references in addition to that of Almeida et al. (2016).
2nd) Some information is missing from the Study area section. For example, the size of catchment area of BV reservoir, plankton composition in the long-term, organic carbon, chlorophyll-a and nutrient data and so on.
3rd) Oxygen undersaturation is reported during mixing for the whole water column. This is a bit surprising. It could reflect that sedimentary organic matter is so high that results in excess oxygen consumption that exceeds net oxygen production by phytoplankton throughout that period. Therefore, some supplementary data on sedimentary organic carbon, nitrogen and phosphorus might aid the reader to understand why this is so.
4th) Novel methodologies also have drawbacks and free-oxygen monitoring is not an exception. I agree with authors’ reluctance for not using it since strong winds make reaeration coefficient estimation unreliable. But the bottle incubation technique for measuring planktonic GPP and R has also limitations and these have been reported many times. Authors must be aware of them and provide a reference citation about.
5th) It is unclear the rationale behind the export capacity of organic matter from biomass settling (Lines 177-178). Please provide a reference and explain it more thoroughly.
6th) Some statistical testing seems advisable in some sentences when making comparisons (“higher than…”, “lower than…”). An example of this neglect can be found in Line 275. Also, the statistical significance of the GPP:R long-term trend is not ascertained (Line 300, Fig. 6), and the rate of increase differs between what is written in the text and the equation shown in the figure.
7th) Lines 317-318. It is doubtful that metabolic rates experience minor variations in time if one takes a look at Fig. 5 and Lines 2-58-268.
8th) The discussion section is a bit cumbersome and could be certainly shortened. For example, there are some obvious explanations that could be deleted, such as the ones in Lines 318-328.
9th) Since data on chlorophyll-a and dissolved organic carbon are lacking, authors fail to make their point stronger when dealing with the vagaries of and GPP limitation by the underwater light climate in Lines 330-337. If you have some data on those variables, you could try to envisage how much GPP variability is explained by such a climate (see Kirk’s book “Light and photosynthesis in aquatic environments”).
10th) Line 364. “Intensity of the winter”. What is it? Please explain it to a non-Mexican audience.
11th) Metabolism data in subtropical and tropical environments are much lower than those in temperate areas. Hence, it is good news to refer to this fact (Lines 433-442). And it would be even better if authors could add a table with most of them (they are not many!), also reporting older studies, for which bottle incubations were used, such as those by William Lewis, John Melack, Jean Pierre Carmouze, Mary Burgis et al. in African and South American lakes in the seventies.
12th) Literature references must be checked. Pages are missing in some instances.
13th) Figure 1 is interesting but the reader would gain more insight if wind- and upper mixed layer plots were added.
14th) Authors have used oxygen isoline as an index of mixed layer (Fig. 3). This is somewhat misleading. They do not always match. I suggest to reconsider such calculations and figure.
15th) Figure 4 is unuseful because it is of minor value within the long-term context addressed in the study.
16th) Tables 1-3. The variability-oriented reader will miss standard deviations for all variables reported.
17th) Please report variables represented by capital letters in Table 3. It is annoying to search for them in the main text when looking at the Table.

---

## Round 0.2 · Minor Revisions

I appreciate your effort reviewing the ms. This new version of your ms has been improved substantially and your study is a very valuable contribution to the field of aquatic ecology. On a scientific level, it is now Acceptable. Congratulations.

[# Before acceptance, a staff check has noticed that the language of the submission needs to be improved. Please can you have a colleague proficient in English go through the manuscript one more time #]

---

## Round 0.3 · accepted · Accept

The English language has been checked satisfactorily and now the manuscript can be accepted to be published in PeerJ. Congratulations again and we expect to receive more of your manuscripts in the future.

#